



# Camera-based Water Stage and Discharge Prediction with Machine Learning

Kenneth W. Chapman[1], Troy E. Gilmore[1], Christian D. Chapman[2], Mehrube Mehrubeoglu[3], and Aaron R. Mittelstet[1]

[1]University of Nebraska Lincoln, Lincoln, Nebraska
[2]MIT Lincoln Laboratory, Lexington, Massachusetts
[3]Texas A&M University Corpus Christi, Corpus Christi, Texas

**Correspondence:** Kenneth W. Chapman (kchapman12@huskers.unl.edu)

**Abstract.** Time-lapse imagery of streams and rivers provide new qualitative insights into hydrologic conditions at stream gauges, especially when site visits are biased toward baseflow or fair-weather conditions. Imagery from fixed, ground-based cameras is also rich in quantitative information that can improve streamflow monitoring. For instance, time-lapse imagery may be valuable for filling data gaps when sensors fail and/or during lapses in funding for monitoring programs. In this
study, we automated the analysis of time-lapse imagery from a single camera at a single location, then built and tested machine learning models using programmatically calculable scalar image features to fill data gaps in stream gauge records. Time-lapse images were taken with a fixed, ground-based camera that is part of a documentary watershed imaging project (https://plattebasintimelapse.com/). Features were extracted from 40,000+ daylight images taken at one-hour intervals from 2012 to 2019. The algorithms removed dawn and dusk images that were too dark for feature extraction. The image features
were merged with United States Geological Survey (USGS) stage and discharge data (i.e., response variables) from the site based on image capture times and USGS timestamps. We then developed a workflow to identify a suitable feature set to build machine learning models with a randomly selected training set of 30% of the images with the remaining 70% for a test set. Predictions were generated from Multi-layer Perceptron (MLP), Random Forest Regression (RFR), and Support Vector Regression (SVR) models. A Kalman filter was applied to the predictions to remove noise. Error metrics were calculated, including
Nash-Sutcliffe Efficiency (NSE), Prediction Bias (PBIAS), RMSE-Standard Deviation Ratio (RSR), and an alternative metric that accounted for seasonal runoff. After suitable features were identified, the dataset was divided into test sets of simulated data gaps for 2015, 2016, and 2017. The training sets for each gap were features from contiguous images and sensor readings before and after the gaps. NSE for the year-long gap predictions ranged from 0.63 to 0.90 for discharge and 0.47 to 0.90 for stage. The predictions for 2015 and 2017 displayed lower prediction errors than for 2016. The 2016 discharge was significantly
higher than training data, which could explain the poorer performance. First and second half-year test sets were created for 2016 along with MLP models from before/after training sets for each of the gaps that held discharge measurements similar to those in the gaps. The half-year gap models' predictions improved NSE, PBIAS and RSR. The results show it is possible to extract features from images taken with the downstream facing camera to build machine learning models that produce accurate stage and discharge predictions. The methods employed should be transferrable to other sites with ground-based cameras.





# 1   Introduction

Accurate measurement and modelling of stream stage and discharge are important for daily water management, flood fore-
casting and management, assessing compliance with water use agreements, and for the design of reservoirs, water supply
systems, bridges, and culverts (Boiten, 2008). Continuous time series data from stream gauges are also critical for calibrating
and/or validating groundwater and surface water models while gaps in these data increase uncertainty in the model predictions.
Stream stage is typically measured with floats, pressure, optical, and acoustic sensors (Turnipseed and Sauer, 2010). These
traditional sensors can fail and require regular maintenance, both of which are costly. As a result, gaps in stream stage and
discharge records may occur due to improper installation (e.g., especially during short-term studies, when site characteristics
are not well-known), equipment failure, and/or gaps in funding for monitoring programs. In this study, we focus on a passive
monitoring approach using time-lapse imagery that can be combined with traditional sensor measurements that is suitable for
filling gaps in streamflow records.

Extraction of discharge and stage from image series and/or video is increasingly common (Wagner and Woodside, 2020), but
widespread adoption of these methods in monitoring networks has yet to occur. Direct measurement of stage with still imagery
(without machine learning models) has been shown to have ±3 mm accuracy in a laboratory setting (Gilmore et al., 2013; Zhang
et al., 2019) and this technology has been applied in field studies (Etheridge et al., 2015). Discharge measurements may also
be made with large scale particle velocimetry (LSPIV) (Muste et al., 2004, 2008, 2011, 2014), space-time image velocimetry
(STIV) (Zhen et al., 2019), optical tracking velocimetry (OTV) (Tauro et al., 2018), and other techniques (Young et al., 2015).
The application of a Kalman filter can help reduce noise in time-series predictions. The efficacy of the use of Kalman filters to
filter hydrological data is also addressed in the literature (Ritter and Muñoz-Carpena, 2013; Jamal and Linker, 2020).

The objective of this research was to fill stream discharge and stage data gaps by programmatically calculating features from
image series using machine learning models for a single location with a single camera to establish the feasibility of extending
these methods to other locations, with different image formats, lighting conditions, etc. Several studies have investigated
methods to fill gaps for hydrological and other environmental data (Ren et al., 2019; Tfwala et al., 2013; Thurstan et al.,
2015; Kim et al., 2019; Dastorani et al., 2009). Machine learning has also been used to predict stream stage and discharge
based on (1) upstream gauging station data and/or water level from other sites (Chen et al., 2020; Gong et al., 2016; Yoon
et al., 2011; Tfwala et al., 2013) (2) other hydrologic data (e.g., precipitation, tide level, etc.; (Chen et al., 2020; Gong et al.,
2016; Yoon et al., 2011)) and (3) improved methods for developing rating curves Jiang et al. (2013), but none of these studies
combined programmatic labelling of images with stream stage and discharge measurements from existing pressure transducers,
calculated image features, machine learning, and Kalman filtering to improve prediction accuracy. Furthermore, the hourly data
time intervals used in this study have higher temporal resolution than the 6-hour to monthly time intervals used in other studies
(Chen et al., 2020; Seo et al., 2018; Gong et al., 2016; Yoon et al., 2011; Tfwala et al., 2013; Jain, 2012). Lastly, previous





studies have not allowed for qualitative assessment of site conditions when data gaps have occurred, whereas our time-lapse imagery provides critical information for qualitatively (visually) evaluating model performance.

In this paper we focused on the benefits of a time-lapse camera co-located with a stream gauging site, especially for filling gaps in streamflow (stage and discharge) records. Key objectives of the project addressed (1) whether time-lapse imagery collected for a documentary project also provide quantitative streamflow information suitable for calibration and/or validation of other hydrologic models (Arnold et al., 2012). We also (2) tested specific cases of full-year and half-year gaps in streamflow data to simulate whether imagery and machine learning could be used to fill in substantial gaps in traditional streamflow monitoring to improve overall stream monitoring programs and other hydrologic studies that rely on time series data from these programs. The study highlights some benefits of passive monitoring at stream gauging sites, including a visual record of the scene for 2-D analysis as opposed to only point measurements from the stage sensor, and the ability to visually assess streamflow conditions during periods when traditional sensor data is unavailable. Future work could also address similar approaches for real-time data validation and/or qualitative assessment of site conditions (e.g., presence of ice or other flow obstructions).

## 2 Study site and data description

Multiple data sets, consisting of stream stage measurements and images, were considered. Options included the GaugeCam project (Gilmore et al., 2013), Platte Basin Timelapse (Forsberg and Farrell, 2011-2020) Mick's Slide and North Platte River State Line Weir camera sites, and other sources. Images and measurements from the North Platte River State Line Weir site were selected due to the high image resolution (4288x2848 RGB), large number of images (57,544), and proximity to a USGS stream gage station (USGS 06674500 NPRSLW station, 2020). The high-resolution daytime images represent a river scene that includes flow over a weir (Figure 2). The water surface appearance varies with the discharge rate. The images were captured for a documentary project, without installation of specific reference points that would aid image-based detection of stream stage. Over the study period, the camera moved roughly ±4° in rotation and ±0.25 m in translation, thus changing the position of the camera and skyline in the images and making direct image measurements more difficult due to calibration issues.

### 2.1 Study site

The studied USGS stream gage is located at the Wyoming-Nebraska state line (USGS 06674500 NPRSLW station, 2020) southeast of Henry, Nebraska. The North Platte River drainage area upstream of the Wyoming-Nebraska State line gage is approximately 57,544 $km^2$ (USGS, 2020). Streamflow at the gage is strongly affected by snowmelt in the headwaters, originating in the Rocky Mountains. Other factors that influence discharge include diversion for irrigation and transbasin water transfer, groundwater withdrawals, return flows, and reservoirs. Mean streamflow at this site for water years 1929 through 2017 was 22.4 $m^3/sec$. For water years 2015, 2016, and 2017, mean discharge was 27.4, 44.9, and 29.0 $m^3/sec$, respectively.

## 2.2 Data description

Data for the North Platte River State Line Weir site came from two sources. The ground-truth stream stage sensor measurements and discharge calculations were downloaded from the USGS (USGS 06674500 NPRSLW data, 2020) for every fifteen minutes from 09-June-2012 to 11-October-2019. Images, taken hourly, were provided by the Platte Basin Timelapse for the same time period. Exposure, shutter speed, capture times, and other data were extracted from each of the image's Exchangeable Image File (EXIF) metadata written into the images at the time of capture.

## 3 Methods and materials

Image-based machine learning models were trained and tested to estimate stream stage and discharge. Although our machine learning models are not run-off models or distributed watershed models (they are based on on-site imagery, trained by same-site gauge data), we evaluated performance primarily on the same error metrics that are ubiquitous in hydrological modelling studies and which allow comparison with other studies. We also considered upper and lower benchmarks to evaluate discharge model performance with simple benchmarks determined from on-site gauge data. The selection of these benchmarks is discussed further in the Section 4.6.

The higher-level process employed in developing and testing programmatically calculable scalar image features was as follows:

1. Select a suitable data set that includes images and accompanying stream stage sensor and discharge measurements.

2. Identify and extract features for each of the images representing stage and discharge.

3. Create machine learning models with features from a training set of images well suited for identification of effective image features.

4. Measure prediction performance of the models for an independent test set of images.

5. Return to step #2 to create new features and repeat until the prediction performance is acceptable.

6. Use the image features to create models to predict measurements for three year-long data gaps.

7. Measure and report the performance ratings for the three, year-long gaps.

It should be noted that the use of machine learning and image processing are at the core of this study and are discussed in detail throughout. However, the focus here is not the inner workings of these tools, but to apply these tools to ascertain stream stage and discharge measurement prediction accuracy. Additionally, we built prediction models rather than performed direct measurements.



### 3.0.1 Training and test sets

The data are separated into two distinct training and test set scenarios. The first scenario focuses on the quality of the image features to facilitate in their development rather than on the machine learning algorithm. In this scenario, the entire data set of 40,000+ daytime images from 2012 to 2019 with features that were calculable was divided into a random training of 13879 images ( 30% of images) and a test of 28181 images ( 70%). We call this the image feature development scenario. Ng (2018) identifies this as the "Dev (development) set — which you use to tune parameters, select features, and make other decisions

regarding the learning algorithm." Ng also notes that "Before the modern era of big data, it was a common rule in machine learning to use a random 70%/30% split to form your training and test sets. This practice can work, but it's a bad idea in more and more applications where the training distribution ... is different from the distribution you ultimately care about ..."

The second category of training and test sets are the use cases (Table 1). Use cases are cases that allowed us to test whether we could fill gaps in the data with training data taken from before and after the gap to be filled. We selected test sets that simulated

one-year gaps for the USGS years of 2015, 2016, and 2017. Those years were selected to limit the scope of the research and still provide a viable test of the models to perform predictions. We then created training sets consisting of sequential leading and trailing data, e.g., 5,000 images before and after the gap to be predicted. It should be noted that additional training sets with different amounts of leading and trailing data (500, 1,000, 2,500, and 8,500) were tested and are available at Kenneth W. Chapman (2020), but the 5,000 before/5,000 after training sets were found to yield satisfactory results.

**Table 1.** Use-case Test and Training Sets. Each line in the table represents three training sets made up of leading and trailing data.

| Gap year | Selection type | Training set before images | Training set after images | Training set total images | Test set images |
|---|---|---|---|---|---|
| 2015 | Sequential | 5,000 | 5,000 | 10,000 | 5,635 |
| 2016 | Sequential | 5,000 | 5,000 | 10,000 | 5,376 |
| 2017 | Sequential | 5,000 | 5,000 | 10,000 | 7,377 |

## 3.1 Feature development

The development of effective image features was achieved through collaboration between the imaging engineer and a hydrologist, merging the expertise of the two disciplines. The identified effective features successfully predicted stream stage and discharge for the test set selected for that scenario and were optimized through an iterative process involving refinement of the programmatically calculated features.

### 3.1.1 Workflow

Figure 1 is a flowchart that shows the image feature development workflow. The individual elements of the workflow are described in more detail in Appendix A, but it is helpful to understand the context of each element within the workflow.

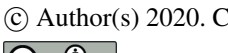



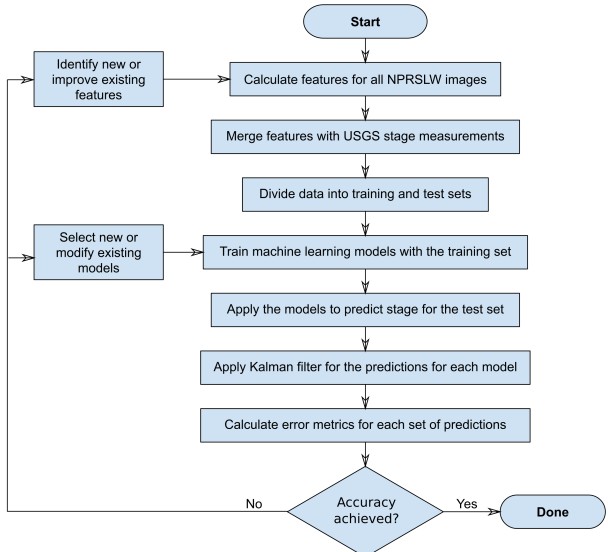

**Figure 1.** Flowchart of the workflow to develop and calculate image features, merge them with point sensor stage measurements, create machine learning classifiers, and measure their prediction accuracy.

### 3.1.2 Image features

Machine learning models based on images often require laborious manual annotation or labelling of images. The large number
of images in the data set and limited resources made it impracticable to perform such manual annotation; therefore, methods to calculate scalar image features were implemented in C++ and a workflow to develop, annotate, refine, and prune features was created (Figure 1). The features were developed with the help of the OpenCV (OpenCV, 2019) computer vision library. Two types of scalar features were extracted: Whole-image features and domain-informed features. Whole image features are generic features such as average pixel intensity, image entropy, etc. that could be extracted from any image. Domain informed
features are ones a hydrologist could identify specific to the domain of hydrology such as the shape and texture of the white water, the colour of the water above and below the weir, etc. These scalar values were stored as tables in Comma Separated Value (CSV) files. Each table row holds the image capture timestamp, camera settings (shutter speed, f setting, ISO speed, etc.), sensor measurement timestamp (closest to the capture–must be $< 15$ minutes or the image is not used), sensor measurements for stage and discharge, and scalar values for each of the calculated image features. These features are available at Kenneth
W. Chapman (2020).

Whole image features represent generic statistics about the image as a whole, irrespective of the domain. These include:

– Intensity mean and sigma - average and standard deviation of the pixel intensities



- Entropy mean and sigma - average and standard deviation of the Shannon entropy and Hartley function calculated from all the pixels within a specified radius of each of pixel position

- Hue, Saturation, Value (HSV) mean and sigma - average and standard deviation of each of the HSV planes which define colour for each pixel with values of Hue, Saturation and Value

It was observed that the shape and texture of the turbulent water (whitewater) below the weir varied with stream stage and discharge (e.g., Muste et al. (2014)). Additionally, colour, pixel intensity, and texture of the water in specific areas above and below the weir also varied with changes in stage and discharge. First, methods to find each weir location in the image were
developed. Then, algorithms were created to measure the features. Figure 2A illustrates a typical image from the data set while Figure 2B shows an image annotated with the results from a weir search. Unlike other studies that used edge detection tools to track the water moving up and down a bridge support or riverbank, there were no convenient reference objects to determine stage in the image scene at the North Platte River State Line Weir.

The first set of domain informed features (Figure 3) consists of scalar measures of the shape, colour and texture of the
turbulence below the weir. This process included the following steps: (1) find the line of the weir, (2) segment the whitewater, and (3) calculate measures based on the segmented whitewater. Whitewater measures include colour and texture within the whitewater area (Shannon entropy, colour (HSV) statistics, and intensity statistics) and shape characteristics (area, perimeter length, and distances of whitewater contour points from the weir (minimum, maximum, mean, and standard deviation of lengths).

The second set of domain informed features are based on the observation that colour, intensity, and texture of the water in specific areas above and below the weir varied with changes in stage and discharge. The Regions of Interest (ROIs) from which these features were calculated are shown in yellow for the area defined above the weir and blue for below the weir (Figure 3). The objective was to measure the difference in appearance between the upstream and downstream water surfaces relative to stage and discharge.

**3.1.3 Feature calculation challenges**

A typical image is shown in Figure 2A. All effective features can be extracted from these types of images. Because there is significant variation in the images from what was typical, each feature could not be calculated on all the images and some images even needed to be eliminated. Over time, the position of the weir in the image varied showing translation and rotation due to drift in location of camera position. This was accommodated by a weir position search algorithm. Edge detection was
performed within a specified ROI and a line fit was calculated from qualifying points. If the angle of the fit line was within a specified minimum and maximum, the weir search was defined as successful. There were 55,804 total images in the data set. Of those, features were calculated for 42,059, which did not have insufficient light (Figure 4E) or occlusions like frost on the lens (Figure 4B). Of the 42,059 processable images, it was possible to find the weir in 36,134 images for which both weir and whole image features could be calculated. The remaining 5,925 images had debris or ice and snow that prevented the weir find,
so only whole image features were calculated.



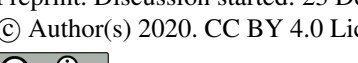



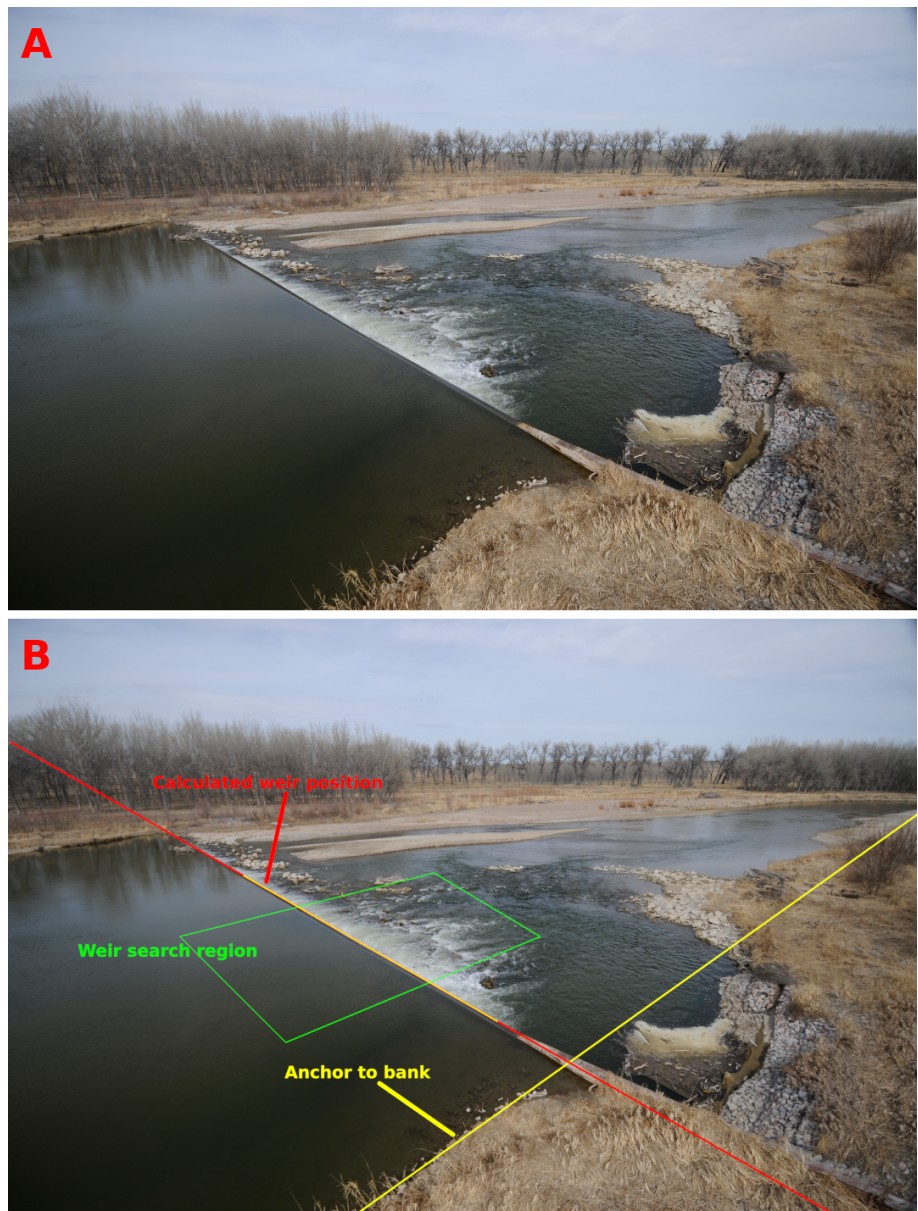

**Figure 2.** A: Typical Platte River Basin image at North Platte River State Line Weir site. B: Same image with annotations to show the weir search region (green), bank location (yellow), and weir found location (red)

At times, the weir could not be found because it was occluded in the images by elevated stage (Figure 4C), e.g., debris (Figure 4D), and/or ice and snow (Figure 4A). This challenge was accommodated by setting the below-weir shape and texture features to an artificially low "not calculable" value. These types of images were included in all the training sets, test sets,





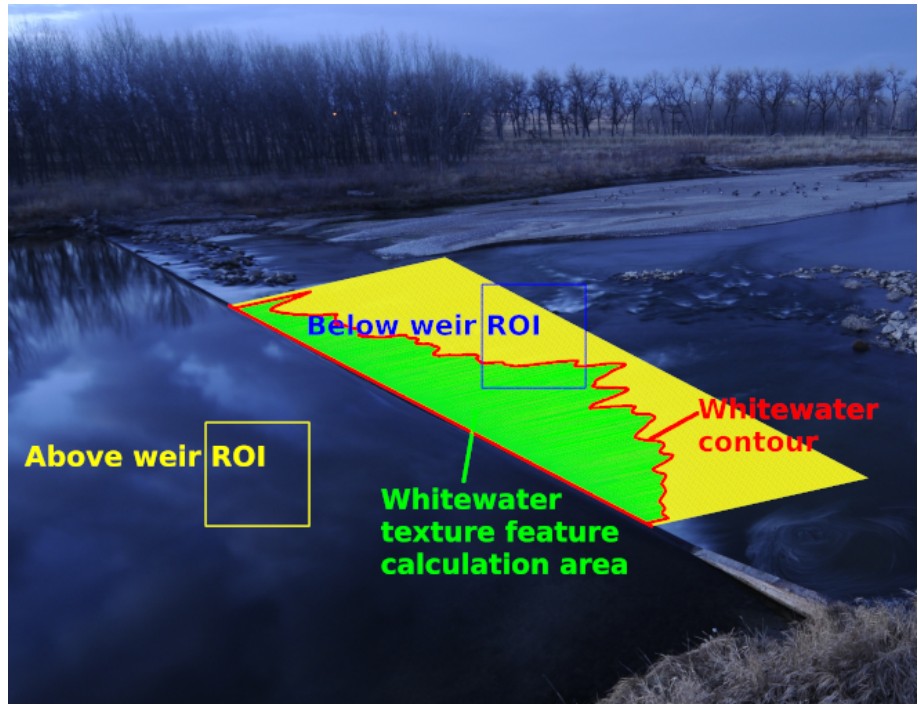

**Figure 3.** North Platte River State Line Weir image with annotation that shows domain informed features including whitewater weir contour, texture, and colour calculation area and above and below weir texture and colour calculation regions. ROI=Region of Interest

and predictions as the other features combined with the knowledge that their weir positions were not findable allowed for the

creation of satisfactory models.

The camera system collected images during daylight hours, but some images were still too dark to extract effective features. These dawn/dusk images were eliminated from evaluation if the mean pixel intensity of the whole image was below a threshold rendering the image too dark. Figure 4E shows a typical dark image. Images with ice on the lens such as Figure 4B present the same kind of problem. These images were also eliminated if whole image contrast was below a minimum threshold.

## 3.2   Models, predictions, noise filtering, and error metrics

### 3.2.1   Models

Three types of models were created for each training set. The models were selected based on three criteria: 1) Suitability to predict scalar stage and discharge measurements from a scalar feature set, 2) availability in standard machine learning tools such as Weka (Frank et al., 2016), SciKit Learn (Buitinck et al., 2013), and R (R Core Team, 2016), and 3) use of the tools in

similar types of machine learning tasks (Jiang et al., 2013; Jain, 2012; Yoon et al., 2011; Gong et al., 2016; Tfwala et al., 2013; Araujo et al., 2011; Seo et al., 2018; Fukami et al., 2020).

The models chosen were


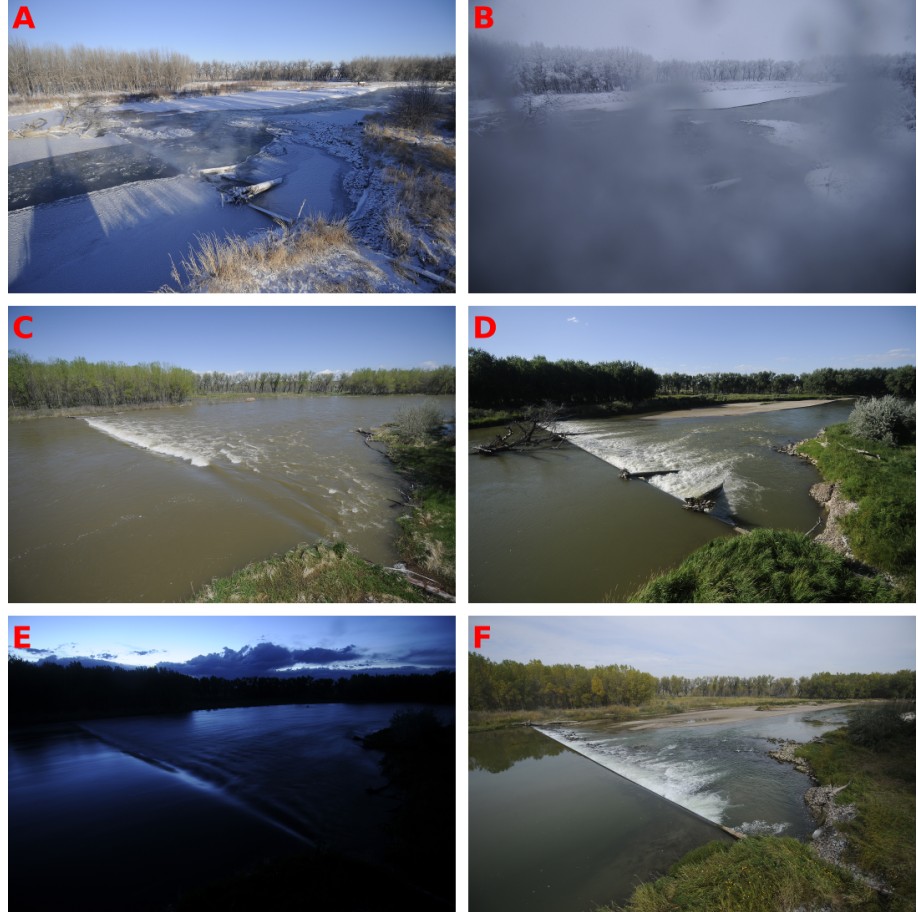

**Figure 4.** Examples of Platte River Basin images where weir position is not found: A) Snow, ice, and debris, B) ice or frost on the lens, C) elevated stage, D) debris occludes parts of the weir, E) dawn/dusk image that is too dark, and F, a typical "good" image.

- Multi-layer Perceptron (MLP) – Artificial Neural Network (see Section 11 of (Hastie et al., 2009))

- Random Forest Regression (RFR) – (see Section 15 of (Hastie et al., 2009))

- Support Vector Regression (SVR) – (see Section 12 of (Hastie et al., 2009))

While it is possible to identify strengths and weaknesses of the different models (Fukami et al., 2020), Ng (2018) emphasizes the iterative nature of machine learning algorithm development. In the spirit of this admonition, error metrics shown in Section 3.2.3 with equations in Appendix B were calculated and minimized through an iterative process to select features and identify a model that best met the objective of the study: creation of predictive models to accurately fill data gaps.

The inputs to the models, predictions, and the models themselves are available for download from Kenneth W. Chapman (2020). These include training and test feature and sensor data in CSV file format, prediction files that hold observed vs.





calculated tabular data, Kalman filter results, and error metrics in Open Document Spreadsheet (ODS) format, and the models themselves in Weka model format. The model creation and prediction processes are well described in "The Weka Workbench" (Frank et al., 2016).

### 3.2.2  Predictions and noise filtering

After the models were created, the test case features were transferred to each of the models to produce predictions, which were then filtered for noise. The true stream stage and discharge of a stream is typically highly correlated over short time intervals. This correlation can be used to reduce prediction error from model variance by applying Kalman filtering (Kalman and Bucy, 1961) on a time series of stream stage and discharge predictions. The use of Kalman filters is not uncommon in hydrology

(Ritter and Muñoz-Carpena, 2013; Sun et al., 2020). For an introduction and practical perspective on Kalman filters, see Kim and Bang (2018). For a simplified example, assume stage and discharge models produce predictions with noise that is additive, Gaussian with zero mean and a fixed variance, and uncorrelated across predictions. Further assume that stage is a Gaussian process that evolves slowly over time. This makes the joint distribution between recent predictions produced by the model and the true stage well-defined. Kalman filtering is a procedure whereby this joint distribution is maintained, and filtered estimates

of the true stage are produced as the expectation of stage, conditioned on observing several of the model's noisy predictions using this modelled prior distribution.

### 3.2.3  Error metrics

Ground truth is taken to be the stage or discharge values in the USGS data tables for the North Platte River State Line Weir. The predictions are the stage or discharge predicted by the various models and filtered for noise with the Kalman filter. The error

metrics are commonly used metrics within hydrology described in (Moriasi et al., 2007) and (Ritter and Muñoz-Carpena, 2013). The equations for the error metrics are shown in Appendix B. These include Mean Absolute Error (MAE[B2]), Mean Square Error (MSE[B3]), Root Mean Square Error (RMSE[B4]), Normalized RMSE (NRMSE[B5]), Prediction Bias (PBIAS[B7]), Nash-Sutcliffe Efficiency (NSE[B8]), and RMSE-observations Standard Deviation Ratio (RSR[B6]). For benchmark-based performance evaluation we took (1) actual USGS measured daily stream discharge as the upper benchmark, and (2) the median of all measurements

for each day of the year from 2009 through 2014 as the lower benchmark. The upper benchmark is idealistic and the lower benchmark simplistic, but this approach does introduce longer-term seasonal runoff patterns into the evaluation of model performance (as opposed to comparison to the mean observed discharge for the modelled time period, i.e., NSE). We recognize that there is no widespread agreement on exact benchmarks to use, and the approach is not common practice in hydrological modelling (Seibert et al., 2018), but we provide these metrics for future comparison with single-site studies.

## 4  Results

The results are divided into six sections. The first three, (1) image feature development, (2) gap year use cases, and (3) 2016 half-year gap use cases describe the actual model predictions and their performance ratings. The fourth section is a comparison





of our results with those from other stage and discharge machine learning model studies. The fifth section compares the performances of the different machine learning models. The sixth section shows the performance metrics of the MLP discharge

model relative to appropriate upper and lower benchmarks.

### 4.1    Image feature development

Models built with the features identified in the image feature development scenario were investigated. Performance of RFR exceeded that of MLP and SVR based on all error metrics (Table 2). The observed vs. predicted graphs for the MLP (Figure 5) demonstrate that the simulated closely predicted the observation data. Maximum absolute error (86.8 $m^3/sec$) occurred when

high discharge was underestimated by the MLP model, but the MLP model tended to overpredict low flows Figure 5. The low-flow errors were relatively small, but cumulatively led to a small and negative PBIAS (-4.71%).

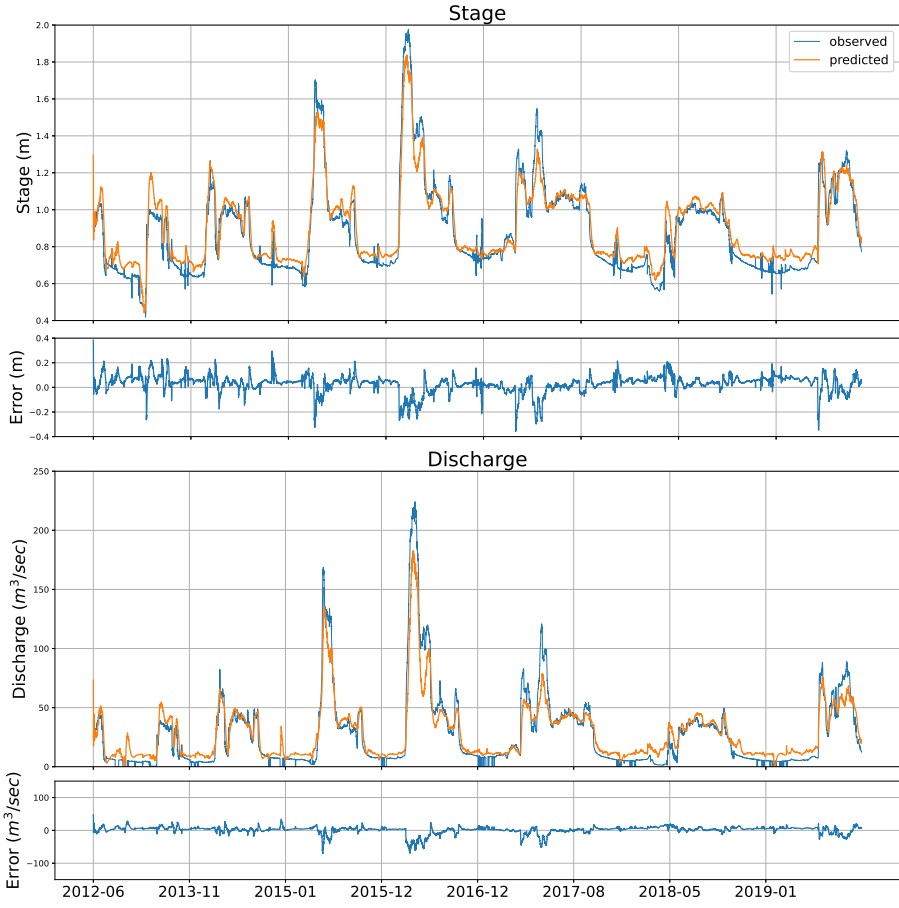

**Figure 5.** Graph of observations vs. predictions for a Multilayer-Perception Model built with a random training set of 30% of the images in the entire data set. The test set consisted of the remaining 70% of the images in the data set. Values are shown only for measurements where usable daytime images were available (nighttime data are excluded).





**Table 2.** Image feature development scenario error metrics for machine learning (ML) models

| Train set | Test set | ML Model | MAE[B2] | MSE[B3] | RMSE[B4] | NRMSE[B5] | PBIAS[B7] | NSE[B8] | RSR[B6] |
|---|---|---|---|---|---|---|---|---|---|
| **Stage** | | | $(m)$ | $(m)$ | $(m)$ | $(m)$ | (% bias) | (SNR) | (RMSE/$\sigma$) |
| 30% | 70% | SVR | 0.08 | 0.02 | 0.14 | 0.16 | 2.02 | 0.67 | 0.33 |
| 30% | 70% | RFR | 0.04 | 0.01 | 0.07 | 0.08 | -0.10 | 0.91 | 0.09 |
| 30% | 70% | MLP | 0.06 | 0.01 | 0.08 | 0.09 | -3.39 | 0.90 | 0.10 |
| **Discharge** | | | $(m^3/sec)$ | $(m^3/sec)$ | $(m^3/sec)$ | $(m^3/sec)$ | (% bias) | (SNR) | (RMSE/$\sigma$) |
| 30% | 70% | SVR | 10.40 | 538.98 | 23.22 | 0.85 | 14.70 | 0.52 | 0.49 |
| 30% | 70% | RFR | 5.64 | 111.22 | 10.55 | 0.39 | -0.78 | 0.90 | 0.10 |
| 30% | 70% | MLP | 7.71 | 132.41 | 11.51 | 0.42 | -4.71 | 0.88 | 0.12 |

We concluded from the image feature development process that the identified scalar image features held adequate information to build machine learning models to predict stage and discharge. We anticipated that they would likely yield good performance ratings for the use cases as well.

## 4.2 Gap year use cases

We created and tested models for the 2015, 2016, and 2017 gap years. Each of the models performed well for 2015 and 2017, but not as well for 2016. The NSE for 2015 and 2017 ranged from 0.83 to 0.90. The predictions for 2016, however, were not optimal and ranged from 0.45 to 0.82. Performance metrics relative to upper and lower benchmarks are described and shown in Section 4.6. Table 3 shows all the error metrics for the year long gaps. Appendix B shows the error metric equations.

The poorer performance of the 2016 model can also be seen in the observed vs. predicted graphs for the MLP model for each year (Figure 6). These results were not due to the feature set because the models for the image feature development scenario accurately predicted both stage and discharge for 2016 (Figure 5). The predicted curves track the observed values closely for 2015 and 2017, but for the times of elevated discharge in 2016, the predicted curve substantially underpredicts the observed discharge. In addition, highest stage and discharge for 2016 ($\approx 240m^3/sec$) were much higher than the most elevated levels for 2015 ($\approx 175m^3/sec$) and 2017 ($\approx 125m^3/sec$).

## 4.3 2016 half-year gap use cases

Additional MLP models were built to understand the poor performance of the 2016 full-year gap models. The observed vs. predicted graphs (Figure 6) illustrated that the 2016 predictions were not satisfactory for stage nor discharge at times of elevated discharge. The stage and discharge from 2015 and 2017 training sets (used to train the models to predict the 2016 gap year) were much lower than the highest stage and discharge in the 2016 data. We suspected that the lack of high discharge in the training data was the source of poor performance in the 2016 gap year model. To verify, we trained half-year gap models for





**Table 3.** Error metrics for machine learning (ML) model types, with training and testing data set size

| Year(s) | Train set | Test set | ML Model | MAE[B2] | MSE[B3] | RMSE[B4] | NRMSE[B5] | PBIAS[B7] | NSE[B8] | RSR[B6] |
|---|---|---|---|---|---|---|---|---|---|---|
| **Stage** | | | | (m) | (m) | (m) | (m) | (% bias) | (SNR) | (RMSE/$\sigma$) |
| 2015 | 10k | 5635 | SVR | 0.06 | 0.01 | 0.10 | 0.11 | -2.48 | 0.87 | 0.13 |
| 2015 | 10k | 5635 | RFR | 0.06 | 0.01 | 0.10 | 0.11 | -4.88 | 0.87 | 0.13 |
| 2015 | 10k | 5635 | MLP | 0.06 | 0.01 | 0.09 | 0.10 | -2.21 | 0.90 | 0.11 |
| 2016 | 10k | 5376 | SVR | 0.13 | 0.05 | 0.23 | 0.22 | 8.83 | 0.63 | 0.37 |
| 2016 | 10k | 5376 | RFR | 0.12 | 0.04 | 0.19 | 0.18 | 5.60 | 0.75 | 0.25 |
| 2016 | 10k | 5376 | MLP | 0.10 | 0.03 | 0.16 | 0.15 | 3.76 | 0.82 | 0.18 |
| 2017 | 10k | 7377 | SVR | 0.05 | 0.01 | 0.08 | 0.08 | 5.85 | 0.83 | 0.17 |
| 2017 | 10k | 7377 | RFR | 0.04 | 0.00 | 0.06 | 0.07 | 4.13 | 0.88 | 0.12 |
| 2017 | 10k | 7377 | MLP | 0.05 | 0.01 | 0.07 | 0.08 | 4.16 | 0.86 | 0.14 |
| **Discharge** | | | | ($m^3/sec$) | ($m^3/sec$) | ($m^3/sec$) | ($m^3/sec$) | (% bias) | (SNR) | (RMSE/$\sigma$) |
| 2015 | 10k | 5635 | SVR | 9.27 | 215.33 | 14.67 | 0.49 | -11.27 | 0.85 | 0.15 |
| 2015 | 10k | 5635 | RFR | 8.82 | 184.25 | 13.57 | 0.45 | -24.15 | 0.87 | 0.13 |
| 2015 | 10k | 5635 | MLP | 8.91 | 151.11 | 12.29 | 0.41 | -6.30 | 0.90 | 0.11 |
| 2016 | 10k | 5376 | SVR | 22.90 | 2043.52 | 45.21 | 0.83 | 37.38 | 0.45 | 0.55 |
| 2016 | 10k | 5376 | RFR | 20.05 | 1319.88 | 36.33 | 0.67 | 26.04 | 0.65 | 0.35 |
| 2016 | 10k | 5376 | MLP | 18.94 | 1387.24 | 37.25 | 0.68 | 32.46 | 0.63 | 0.37 |
| 2017 | 10k | 7377 | SVR | 5.68 | 92.91 | 9.64 | 0.32 | 14.76 | 0.85 | 0.15 |
| 2017 | 10k | 7377 | RFR | 4.54 | 65.80 | 8.11 | 0.27 | -0.39 | 0.89 | 0.11 |
| 2017 | 10k | 7377 | MLP | 4.99 | 61.69 | 7.85 | 0.26 | 1.97 | 0.90 | 0.10 |

the first and second halves of 2016 with surrounding data that included the higher stage and discharge measurements from that same year. NSE improved to a range between 0.87 to 0.95 and PBIAS was reduced substantially compared to the 2016 full-year gap (<18% versus 32%, Table 4). It can also be seen that the predicted curve more closely tracks the observed curve

275  (Figure 7).

## 4.4  Model comparisons

For this study, we used machine learning models (SVR, RFR, and MLP) readily available within common machine learning tools such as Weka, SciKit Learn, and R. These models produced satisfactory results. In general, the MLP (Figure 8; blue bars) demonstrated the highest performance, with the RFR model (orange bars) a close second. The SVR models performed

280  reasonably well, but not as well as MLP and RFR. As expected, there is a noticeable drop in performance for most of the metrics for 2016 gap year due to the training and test set discrepancies for that year.

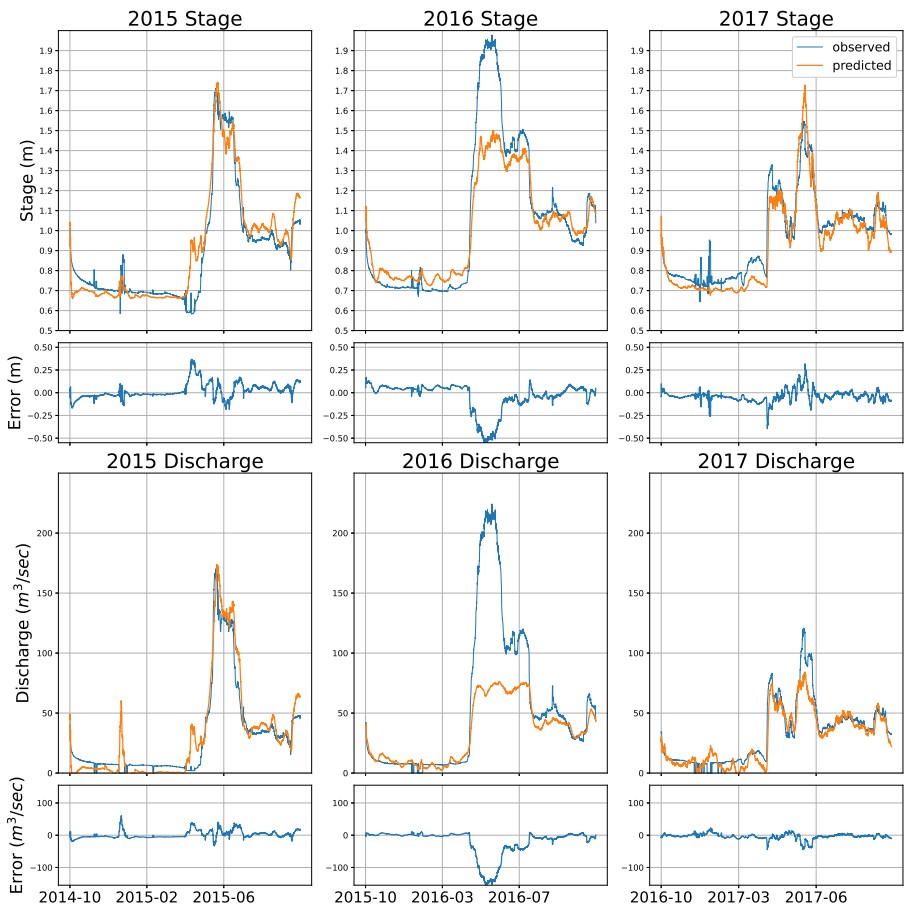

**Figure 6.** full-year observed vs predicted graphs for 2015, 2016, and 2017. Values are shown only for measurements where usable images were available (i.e., nighttime data are excluded).

We conclude that the models, particularly MLP and RFR, performed well and can be used in stage and discharge predictions. There is still an opportunity for research into model tuning and optimization that could improve the reported models' performance by improving prediction results in the case of challenging scenes representing extreme events. It is our intuition that it would be possible to improve the SVR through adjustment to the various parameters within Weka that control the model building, e.g. the complexity parameter, the kernel to use, the tolerance parameter used for checking stopping criterion, etc., but this was beyond the scope of this study.

## 4.5 Comparative studies

Several studies have used machine learning models to predict stream stage and discharge. The reported statistical metrics were RMSE and NSE for all but one study. As can be seen in Table 5, each study reported good results in terms of both RMSE and





**Table 4.** 2016 Half-Year Gap Error Metrics

| Year | Train set | Test set | ML Model | MAE[B2] | MSE[B3] | RMSE[B4] | NRMSE[B5] | PBIAS[B7] | NSE[B8] | RSR[B6] |
|---|---|---|---|---|---|---|---|---|---|---|
| **Stage** | | | | (cm) | (cm) | (cm) | (cm) | (% bias) | (SNR) | (RMSE/$\sigma$) |
| 2016 full-year | 10k | 5635 | MLP | 0.10 | 0.03 | 0.16 | 0.15 | 3.76 | 0.82 | 0.18 |
| 2016 1st half | 10k | 2116 | MLP | 0.07 | 0.02 | 0.12 | 0.13 | 3.37 | 0.91 | 0.09 |
| 2016 2nd half | 10k | 3366 | MLP | 0.55 | 0.01 | 0.07 | 0.06 | 4.37 | 0.95 | 0.05 |
| **Discharge** | | | | ($m^3/sec$) | ($m^3/sec$) | ($m^3/sec$) | ($m^3/sec$) | (% bias) | (SNR) | (RMSE/$\sigma$) |
| 2016 full-year | 10k | 5376 | MLP | 18.94 | 1387.24 | 37.25 | 0.68 | 32.46 | 0.63 | 0.37 |
| 2016 1st half | 10k | 2116 | MLP | 10.43 | 392.81 | 19.82 | 0.50 | 17.96 | 0.91 | 0.09 |
| 2016 2nd half | 10k | 3366 | MLP | 14.13 | 374.24 | 19.35 | 0.31 | 17.49 | 0.87 | 0.13 |

NSE for the domains within which they were working and the goals they set. There were three big differences that sets our study apart from the others:

First, our models included 45 features while the other studies used only one to four features. The large image feature set used in the study produced strong models. Because image features are programmatically calculable, they are easy to merge with ground truth data. They are also easily merged with non-image features like those used in the comparison studies. Use of such additional features potentially increases the accuracy of any models based on images (e.g., precipitation, gauge data from upstream or downstream).

Second, the time interval between feature measurements was hourly for this study while the other studies ranged from six hours to one month. We focused on hourly data in part because images were collected hourly, but also to reconstruct (predict) the maximum amount of gauge data possible. Thus, this study provided more fine-grained time-correlated predictions due to the shorter hourly time interval compared to other studies, thus preserving more information. A drawback to image feature in this study, however, is the lack of nighttime imagery and the occasional artifacts that prohibited analysis of image features.

Finally, this study and two of the compared studies included only on-site data while the other studies used a combination of on-site and off-site data. An example of using off-site data is Tfwala et al. (2013)), where "The idea is to model flow at one gauge as a function of flow at another gauge or gauges." There are trade-offs between these approaches. In some cases, off-site data such as precipitation elsewhere in the watershed can add information to models, but those data may not be well-correlated with stream stage or discharge at the study site. For instance, spatial variability in precipitation could in some cases add noise to the model inputs. Similarly, groundwater and tributary inflows and managed reservoir releases may confound relationships between upstream streamflow data and the given study site. However, in some cases, upstream and downstream discharge can be strongly correlated, if not linearly. These issues could be further explored by incorporating image features from one or more sites and/or incorporating other off-site data in future studies.



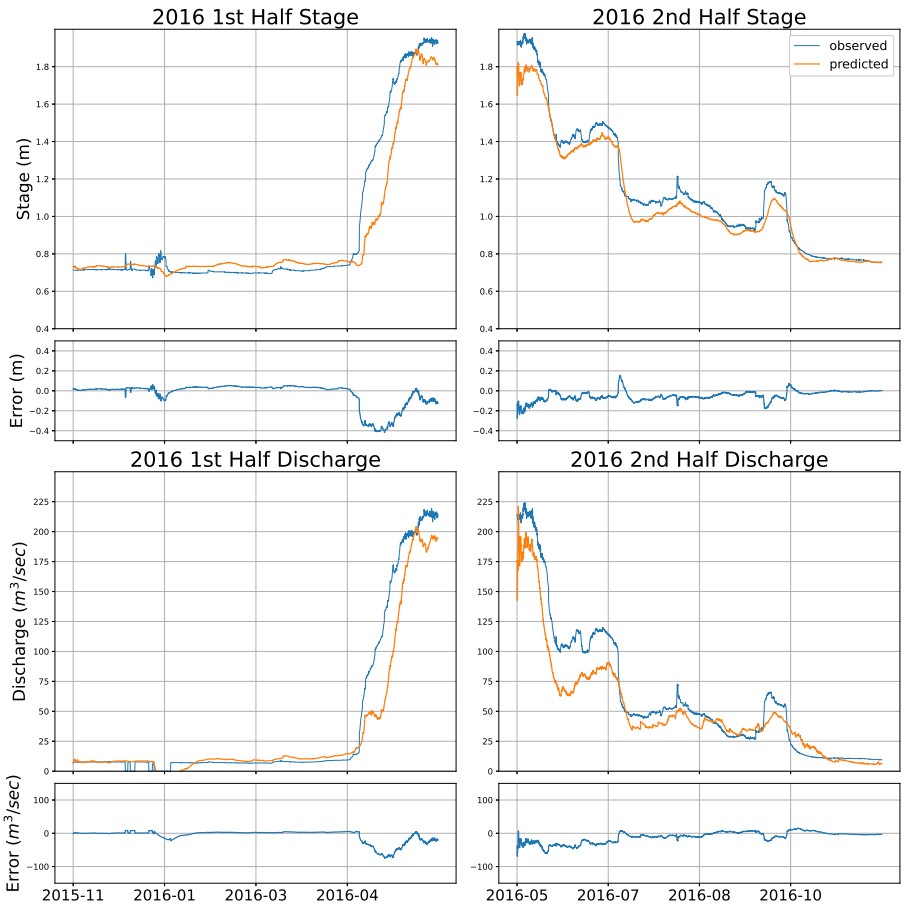

**Figure 7.** Half-year observed vs. predicted graphs for 2016. Values are shown only for measurements where usable daytime images were available (nighttime data are excluded).

We conclude from this comparative analysis that we have been able to build accurate models with images that provide a smaller time interval between predictions, more features than have been available in previous studies and with only on-site data sources. We also believe the model in this study could benefit from the use of features identified in the comparative studies as much as they could benefit from the use of image features when images are available.

### 4.6 Model performance based on daily data

Having established the feasibility of using daytime image features from high-quality images to predict available hourly discharge values over year-long data gaps, we further evaluated model performance on a daily time-step, which is common for hydrologic model calibration (Figure 9). Model performance was evaluated based on the same error metrics as hourly models. We also relied on simple benchmarks (Figure 9) to evaluate model performance relative to longer-term seasonal run-off patterns



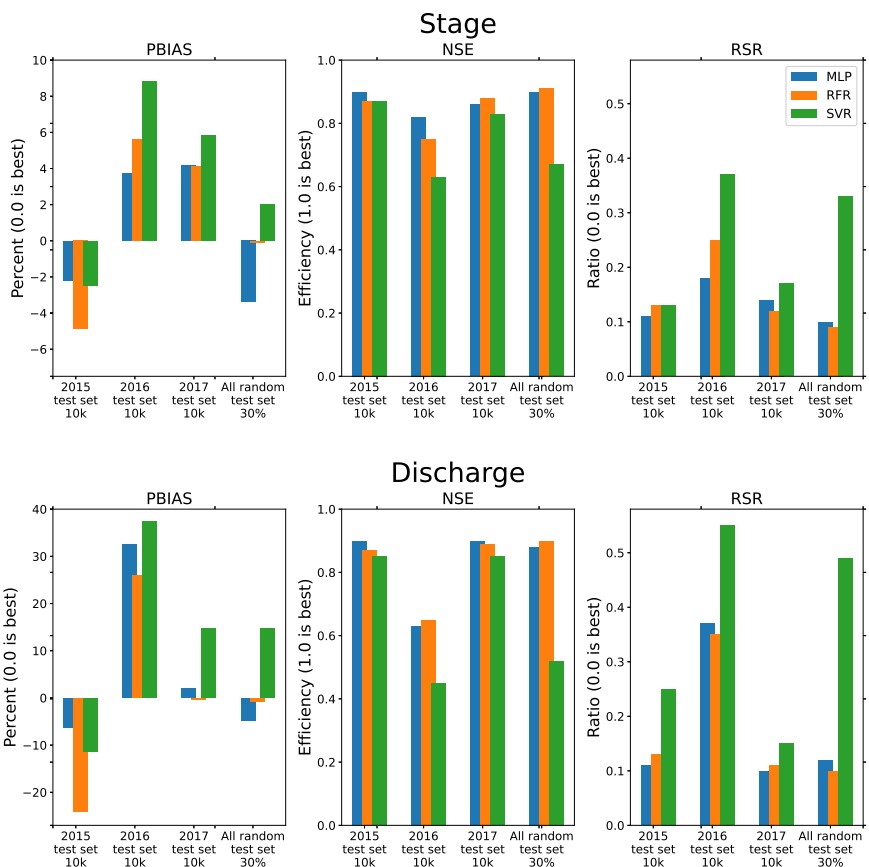

**Figure 8.** PBIAS, NSE, and RSR comparisons for each model type and test set.

at the North Platte River State Line Weir. Collectively, this analysis also tested is hourly daytime-only imagery is representative of daily streamflow.

Based on daily average predictions, MLP model performance (Table 6) was similar to hourly predictions (Table 3). PBIAS for the daily time-step model was greater for daily values; however, still below 20% in the worst case (2016 gap year, -19.4%).

Establishment of a lower benchmark required a method to fill year-long gaps of daily streamflow measurements for each prediction year. Methods that are able to fill gaps of that length depend on off-site data and/or were more suited to fill much shorter data gaps (Dery et al., 2005; Yannick, 2014; Zhang and Post, 2018; Dembélé et al., 2018). Therefore, we decided to use six-year daily median of the USGS measured stream discharge from 2009 to 2014 as the lower benchmark. The lower benchmark therefore reflects long-term seasonal runoff characteristics for the site. For the upper benchmark we used the daily

mean discharge for 2016-2017 (USGS 06674500 NPRSLW station, 2020). This was the same data used to train the machine learning models. It should be noted that uncertainty in discharge measurements range from 5% to 10% as suggested in Tables 4.20 and 4.21 of Boiten (2008) and illustrated in Figure 9. There are discontinuities in both the USGS measured stream





**Table 5.** Comparative Studies

| Study | Time interval | Feature count | Inputs | RMSE range | NSE range | RSR range | PBIAS range |
|---|---|---|---|---|---|---|---|
| **Stage** | | | | *meters* | | | *meters* |
| Chapman et al. (2020)[1] | Hour | 45 | On-site[2] | 0.06 to 0.23 | 0.63 to 0.90 | 0.11 to 0.37 | -4.88 to 8.83 |
| Chen et al. (2020) | Day | 4 | Off-site[3] | 1.12 to 1.71 | 0.65 to 0.71 | N/A | N/A |
| Seo et al. (2018) | Day | 1 | On-site[4] | 0.00 to 0.04 | 0.97 to 1.00 | N/A | N/A |
| Gong et al. (2016) | Month | 4 | On/Off-site[5] | 0.59 to 1.59 | 0.06 to 0.69 | N/A | N/A |
| Yoon et al. (2011) | Six hours | 3 | On/Off-site[6] | 0.17 to 0.19 | 0.53 to 0.63 | N/A | N/A |
| **Discharge** | | | | $m^3/sec$ | | | $m^3/sec$ |
| Chapman et al. (2020)[1] | Hour | 45 | On-site[1] | 7.85 to 45.21 | 0.45 to 0.90 | 0.10 to 0.37 | -24.15 to 37.38 |
| Chen et al. (2020) | Day | 4 | Off-site[3] | 13.54 to 19.56 | 0.54 to 0.83 | N/A | N/A |
| Tfwala et al. (2013) | Day | 3 | Off-site[7] | 124.71 to 150.36 | 0.97 to 0.98 | N/A | N/A |
| Jain (2012) | Day | 1 | On-site[8] | N/A | 0.77 to 1.00 | N/A | N/A |

[1]This study (One year gap predictions only)

[2]Data: Images, Training set: 5,000 before and 5,000 after gap, Test set: All gap year images – 5643 to 7377 images

[3]Data: Pumping rates, recharge rates, discharge from two other stations, Training set: Data from 1986-2008, Test set: Data from 2009-2010

[4]Data: Lagging stage from same site, Training set: Stage measurements from 2009-2014, Test set: 2015-2016

[5]Data: Precipitation, temperature, lagging stage from same site, nearby lake level, Training set: Data from 1998-2007, Test set: Data from 2008-2009

[6]Data: Precipitation, tide level, lagging stage from same site, Training sets: 06/04-08/04, 05/05-11/05, Test sets: 11/04-12/04, 05/20-11/06

[7]Data: Discharge from three other sites, Training set: 1997-2009 (70%), Test set: Data from 1997-2009 (10%)

[8]Data: Lagging discharge from same site, Training set: 2004-2005, Test set: 2006

discharge daily means and in the prediction curves. These discontinuities and the selected benchmarks are addressed further in Section 5.

To incorporate the longer-term seasonal patterns of discharge into model evaluation, results in Table 6 were used in Equation 1 of Seibert et al. (2018)

$$R_{relative} := \frac{R_x - R_{lower}}{R_{upper} - R_{lower}} \tag{1}$$

where $R_x$ is the prediction performance measure, $R_{lower}$ is the lower benchmark performance measure, and $R_{upper}$ is the upper benchmark performance measure.

The benchmark-based performance metrics (Table 7) show the MLP models for all three years performed better than the lower benchmark except for PBIAS for the 2016 gap year. In addition, they show the 2015 and 2017 models performed better than the 2016 model, as expected.

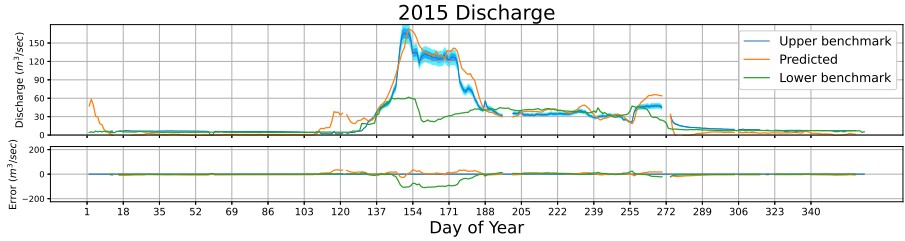

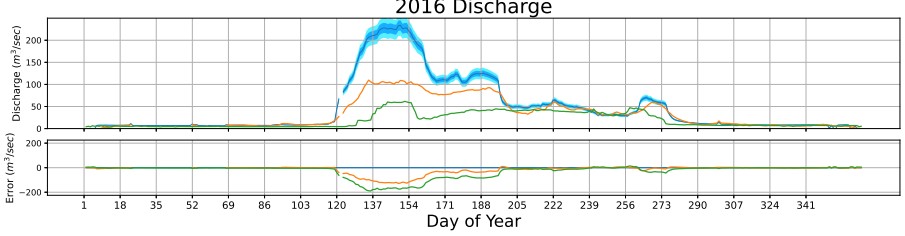

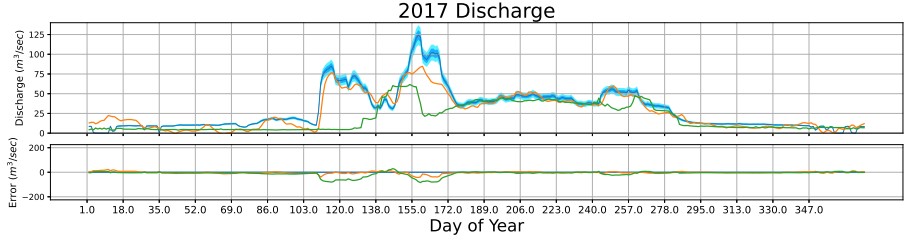

**Figure 9.** Benchmark comparisons for 2015, 2016, and 2017 gap years. The upper benchmarks are the blue lines which show the daily means of the USGS stream flow measurements with light blue shaded areas around them that show the 5% and 10% error bands as discussed in Boiten (2008). The lower benchmarks are the green lines which show the six-year daily median (2009-2014) of the USGS stream flow previous to the gap being predicted. The prediction curves created from the MLP machine learning models are the orange lines.

**Table 6.** Error Metrics for MLP Daily Discharge Model and 6-year Median (2009-2014) Daily Discharge Relative to Observed Data

| Model | MAE[B2] $(m^3/sec)$ | MSE[B3] $(m^3/sec)$ | RMSE[B4] $(m^3/sec)$ | NRMSE[B5] $(m^3/sec)$ | PBIAS[B7] (% bias) | NSE[B8] (SNR) | RSR[B6] (RMSE/$\sigma$) |
|---|---|---|---|---|---|---|---|
| Lower benchmark | 12.48 | 828.28 | 28.78 | 1.07 | 29.18 | 0.38 | 0.63 |
| 2015 MLP Discharge | 8.88 | 152.45 | 12.35 | 0.46 | -6.25 | 0.93 | 0.08 |
| Lower benchmark | 29.50 | 3405.67 | 58.36 | 1.24 | 60.46 | 0.10 | 0.90 |
| 2016 MLP Discharge | 17.05 | 1261.91 | 35.52 | 0.75 | 33.32 | 0.79 | 0.21 |
| Lower benchmark | 12.67 | 525.55 | 22.93 | 0.77 | 36.50 | 0.22 | 0.78 |
| 2017 MLP Discharge | 6.40 | 90.40 | 9.54 | 0.32 | 10.96 | 0.94 | 0.06 |





**Table 7.** Relative performance based on Lower/Upper Benchmarks

| Model | MAE[B2] | MSE[B3] | RMSE[B4] | NRMSE[B5] | PBIAS[B7] | NSE[B8] | RSR[B6] |
|---|---|---|---|---|---|---|---|
| 2015 MLP Discharge | 0.29 | 0.82 | 0.57 | 0.57 | 1.21 | 0.88 | 0.88 |
| 2016 MLP Discharge | 0.42 | 0.63 | 0.39 | 0.39 | 0.45 | 0.77 | 0.77 |
| 2017 MLP Discharge | 0.50 | 0.83 | 0.59 | 0.59 | 0.70 | 0.93 | 0.93 |

## 5    Discussion

The precision of the models depended greatly on the quality of the calculated image features. Based on the image feature
development process, we believe it is possible to find additional image features and make improvements to the current feature
set to improve the precision of the models. Also, future work could be done to improve precision by tuning the machine
learning model parameters. The results showed the RFR and MLP models performed better than the SVR model, although all
models produced reasonable results. As a corollary to that, the most likely reason the 2016 models performed poorer is that
they were trained with features from images that did not fully represent the ranges of stage and discharge manifested in the test
set images. The scope of the research was extended to test that hypothesis as described, but more could be done in that regard
including addition of image features and the use of differential rather than absolute feature measurements.

   This study was focused on the feasibility of a new approach to discharge time series gap-filling (as opposed to run-off or
distributed watershed models) using on-site data. In addition to standard calculations of model error, we also calculated error
using simple upper and lower benchmarks based on on-site data to evaluate performance. Additional exploration of suitable
benchmarks could be done, especially since the inputs to the models in this study are images. There are strategies that could be
employed to create reasonable random variation in those inputs including introduction of Gaussian noise, randomly resizing
the images, adding random spatial variation, etc., but the computational resource to calculate the image features, create the
models, and calculate the predictions made this option impracticable within the scope of this study. We used the USGS sensor
data as the upper benchmark for evaluating model results on a daily time-step. This was the same data used as "ground truth"
to build the machine learning models but there is measurement error associated with these measurements, probably in the 5%
to 10% range (Boiten, 2008). For the lower benchmark, we used the median of all the available data for each day of the year
from 2009 to 2014. By using median daily values from the years immediately previous to the gap years for which the models
were built, the lower benchmark reflected longer-term seasonal patterns of discharge. These are simple benchmarks, but could
be easily replicated for future studies.
Discontinuities in model predictions Figure 9 were a result of images that could not be processed due to ice and debris in the
river and due to fog (Figure 4). Additional research could be performed on ways to ameliorate the problem with images that
were not processable, which is a drawback to any image-based approach. On the other hand, the availability of imagery for gap
filling is beneficial for quickly reviewing image scenes and confirming site conditions, which is not possible with methods that





rely only on statistical methods for gap filling. For instance, imagery for the 2016 gap year clearly shows high flow conditions,
highlighting both the qualitative and quantitative value of image-based approaches.

The studied river scene included a weir that induced turbulence on the downstream side of the weir. We were surprised to learn that the non-weir features were a positive strong contributor to the quality of the models, possibly on par in some cases with the turbulent water (whitewater) feature. Even more surprising was that the non-weir features contributed to prediction improvements even when the weir features were not available. We believe the addition of features based on well-known image
processing algorithms such as edge direction and magnitude in and around the water line, region-based Fourier response, region-based moments and gradients of convolutions, statistical change detection and a variety of additional relative well-known and well understood image features available in OpenCV (2019) and textbooks such as Ballard and Brown (1982) and Gonzalez and Woods (2018) could improve model quality. These and other feature extraction algorithms do not have to be only applied to 2D images, but are equally applicable to 3D (from stereoscopic, laser triangulation, LIDAR, etc.), hyper-
spectral, ultrasonic, and other images. We believe that when image features can be combined with non-image features such as precipitation, cloud cover, temperature, humidity, chemical signature, irrigation, and other measures of local activity, the model performance could show substantial improvements. All of these items are opportunities for future research.

## 6 Conclusions

In addition to identification of features that can predict stage and discharge accurately in straight-forward scenarios for the
selected study site, it is important to assure the data in the training set used to build a model includes the range of values represented in the test set. If the model is used to predict stream stage and discharge outside values of the trained data set, uncertainty in model predictions increases. Although algorithms can be used to ensure that the maximum and minimum observations are included in the training dataset (Nelson et al., 2018), when there is an actual gap (unlike the simulated gaps in Nelson et al. (2018) and other studies) in streamflow records, there is no guarantee that stage or discharge has not exceeded the bounds
of existing observations. In this sense, image-based gap filling has an advantage over purely numerical gap filling methods. Images from the test set (the gap) can be compared to images from the training set to get a sense for whether the training set is adequate to build a good model to fill the gap. In addition, performance relative to the upper and lower benchmarks can help identify inadequacies in the training set. Whichever avenues are pursued in future studies, we believe the use of commonly and inexpensively available software libraries and applications, as used in this study, is important for making machine learning
tools that are more accessible to managers, educators, scientists, and engineers (Saia et al., 2020).

Future research could include testing the efficacy of image-based machine learning models for different sites, improvements in selection of lower and upper performance benchmarks, refinement of machine learning model parameters, and computational improvements to speed and quality of image feature calculation and machine model building. In addition, further research could address similar approaches for real-time data validation and/or qualitative assessment of site conditions (e.g., presence of ice
or other flow obstructions).

*Data availability.* The feature and prediction datasets generated and/or analysed during the current study are available during the review process at Kenneth W. Chapman (2020), but will be made available via the University of Nebraska – Lincoln Data Depository on article acceptance. Contact the Platte River Timeplapse Project to request access to the images used in this study. Contact the USGS to request access to the stage and discharge measurement data used in this study.

*Code availability.* The Weka models generated during the current study are available during the review process at Kenneth W. Chapman (2020), but will be made available via the University of Nebraska – Lincoln Data Depository on article acceptance.

*Video supplement.* Animations of different aspects of the image feature development created in the form of GIF and AVI files during the current study are available during the review process at Kenneth W. Chapman (2020), but will be made available via the University of Nebraska – Lincoln Data Depository on article acceptance.

*Author contributions.* TG and KC were responsible for conceptualization. KC and CC developed the image processing and model code and KC performed formal analysis. KC prepared the manuscript with contributions from all co-authors. TG was responsible for project administration and funding acquisition.

*Competing interests.* No competing interests are present.

*Financial support.* U.S. Department of Agriculture—National Institute of Food and Agriculture NEB-21-177 (Hatch Project 1015698).

*Disclaimer.* Christian D. Chapman is currently an MIT Lincoln Laboratory employee. No laboratory funding or resources were used to produce the result/findings reported in this publication.

*Acknowledgements.* The authors gratefully acknowledge the Platte Basin Timelapse project for providing the images and the United States Geological Survey, Wyoming office for supplying the stage and discharge data.





## Appendix A: Software and procedures

The following procedures and programs were designed to create and test the water stage prediction machine learning models. Each procedure and program is associated, from top to bottom, with the rectangular boxes of the flowchart in Figure 1.

– C++ program calculates scalar features in images of water scenes well suited to inform the creation of machine learning models to measure water stage and write them along with image capture date and time to a CSV file

– C++ program reads USGS sensor data files and merges stage and discharge measurements with the image feature data,
assigning measurements to each image feature set by date and time

– C++ program divides the merged image feature/measurement sets into training and test sets based on user specified percentages

– Procedure to create ANN, Random Forest Regression, and Support Vector Regression models with Weka ((Frank et al., 2016)) from the training sets to predict stage

– Procedure to test the machine learning models with the test sets

– C++ program to apply a Kalman filter to the test set results to refine the predictions

– Procedure to calculate a spreadsheet that holds the predictions and error metrics defined in Moriasi et al. (2007)

– Procedure to develop, evaluate, and refine scalar image features used to build the machine learning models





## Appendix B: Error metric equations

Except fo Nash-Sutcliffe efficiency $\rho_{\mathsf{NSE}}$, all the quantities defined below are distortion measures and lower expected values for them indicate better estimator performance. Nash-Sutcliffe efficiency $\rho_{\mathsf{NSE}}$ has a maximum expected value of 1 and for it higher expected values indicate better estimator performance. Several of the following definitions also depend on the observed sample mean $\bar{y}$:

$$\bar{y} := \frac{1}{n}\sum_{i=1}^{n} y_i^{observed}. \tag{B1}$$

– **Mean Absolute Error (MAE)**

$$d_{\mathsf{MAE}} := \frac{1}{n}\sum_{i=1}^{n}\left| y_i^{observed} - y_i^{predicted}\right| \tag{B2}$$

   – **Mean Square Error (MSE)**

$$d_{\mathsf{MSE}} := \frac{1}{n}\sum_{i=1}^{n}\left( y_i^{observed} - y_i^{predicted}\right)^2 \tag{B3}$$

   – **Root Mean Square Error (RMSE)**

$$d_{\mathsf{RMSE}} := \sqrt{\frac{1}{n}\sum_{i=1}^{n}\left( y^{predicted} - y_i^{observed}\right)^2} \tag{B4}$$

   – **Normalized RMSE (NRMSE)**

$$d_{\mathsf{NRMSE}} := \frac{d_{\mathsf{RMSE}}}{\bar{y}} \tag{B5}$$

   – **RMSE-observations Standard Deviation Ratio (RSR)**

$$d_{\mathsf{RSR}} := \frac{d_{\mathsf{RSME}}}{\sqrt{\frac{1}{n}\sum_{i=1}^{n}(y_i^{observed} - \bar{y})^2}} \tag{B6}$$

– **Prediction Bias (PBIAS)**

$$d_{\mathsf{PBIAS}} := 100 \cdot \frac{\sum_{i=1}^{n}\left( y_i^{observed} - y_i^{predicted}\right)}{\sum_{i=1}^{n} y_i^{observed}} \tag{B7}$$

   – **Nash-Sutcliffe Efficiency (NSE)**

$$\rho_{\mathsf{NSE}} := 1 - \left[ \frac{\sum_{i=1}^{n}\left( y_i^{observed} - y_i^{predicted}\right)^2}{\sum_{i=1}^{n}\left( \bar{y} - y_i^{observed}\right)^2} \right] \tag{B8}$$



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
