# Peer review of "Camera-based Water Stage and Discharge Prediction with Machine Learning"

_Hydrology and Earth System Sciences, 2020_

## Referee Comment (RC1) · Anonymous Referee #1 · 1 Feb 2021

The authors have presented a novel approach for estimation of river water stage and discharge by using camera data, processed through machine learning. This is a potentially useful approach and it may prove to be a viable alternative to the currently used approaches for filling data gaps. The calibration and validation results obtained by using various techniques such as ANN, SVR, etc. are also quite encouraging.

I have some observations from hydrology point-of-view. • What are the extra benefits of using this approach vis-à-vis use of conventional hydrologic methods (such as time series analysis, hydrologic models, use of rating curve to convert river stage to discharge, etc.) of filling data gaps ? As noted, there may be data gaps due to several reasons including malfunctioning of sensors. Likewise, a camera may also malfunction at times. • How this approach is likely to work for rivers with smooth/rough water

surface and the rivers with high sediment load/floating debris ? • How useful will this approach be in night time ? Will it help if a flash device is used in the night time ? The applicability of the technique for the catchment with short response time will be limited if night time photos are not useful. • Authors may please explain how this technique would work for wide rivers and for rivers where the flow passes through multiple channels whose dimensions that keep on evolving and changing ? • • Equation (B4), please add subscript 'i' to y-predicted

---

## Author Comment (AC1) · 6 Feb 2021

(AC) Thank you for your encouraging words and thoughtful questions. We believe these are important comments and questions. We have taken the liberty of splitting them into a numbered list to more easily address them. Our responses are in red text and marked as "AC".

The authors have presented a novel approach for estimation of river water stage and discharge by using camera data, processed through machine learning. This is a potentially useful approach and it may prove to be a viable alternative to the currently used approaches for filling data gaps. The calibration and validation results obtained by using various techniques such as ANN, SVR, etc. are also quite encouraging. I

have some observations from hydrology point-of-view.

1. What are the extra benefits of using this approach vis-à-vis use of conventional hydrologic methods (such as time series analysis, hydrologic models, use of rating curve to convert river stage to discharge, etc.) of filling data gaps?

(AC) Imagery gives insight to site conditions that no other gap-filling method can provide, as mentioned in the abstract and introduction to the paper. In the case of a real gap in stream gauge data, no other method can provide a qualitative on-site assessment to confirm that the gap filling method reasonably represents the actual streamflow during that time period. Imagery also provides important context (e.g., presence of ice or other obstructions, site conditions when a technician cannot be present). We will revise the manuscript to more further emphasize these comparisons with other methods. (anticipated length: 2-3 sentences)

2. As noted, there may be data gaps due to several reasons including malfunctioning of sensors. Likewise, a camera may also malfunction at times.

(AC) This is true. We will add a note to this effect in the paper. (anticipated length: 1-2 sentences)

3. How this approach is likely to work for rivers with smooth/rough water surface and the rivers with high sediment load/floating debris?

(AC) These are opportunities for future studies. This is addressed further in the answer to question #5.

4. How useful will this approach be in night time approach be in night time ? Will it help if a flash device is used in the night time? The applicability of the technique for the catchment with short response time will be limited if night time photos are not useful.

(AC) When there are no nighttime images, there is no way to make predictions with only images. That being said, we successfully measured water level with cameras using IR LED's for nighttime images in a previous study (Gilmore, T. E.,

Birgand, F., and Chapman, K. W.: Source and magnitude of error in an inexpensive image-based water level measurement system, Journal of Hydrology, 496, 178–186, https://digitalcommons.unl.edu/cgi/viewcontent.cgi?article=1769context=natrespapers, 2013.). Of course, this is also an opportunity for future studies, but given our previous experience, we believe there is optimism for success. We will note this in the section that discusses night time images. We will also note that a lack of night time imagery could be more important in flashier hydrologic systems. (anticipated length: 2-4 sentences)

5. Authors may please explain how this technique would work for wide rivers and for rivers where the flow passes through multiple channels whose dimensions that keep on evolving and changing?

(AC) We believe these are important questions that are opportunities for future studies. The process defined in this paper that helped us identify image features to produce precise machine learning models was an outcome of this paper. The use of that process to investigate different scenarios was beyond the scope of this paper, but is integral to our ongoing research. The development of precise models in these different scenarios will almost certainly require the development of additional image features, but it also might require the introduction of additional cameras with varying views of the scene as well as leading and lagging time series data from upstream and downstream sensors and cameras, weather data, etc. We are in complete agreement with the reviewer that these are important questions, but beyond the scope of the current study. We will provide more discussion of this issue, including the relative strength of image-based methods (which might actually reveal the degree to which channel dimensions evolve – something not possible with a traditional pressure sensor) in capturing these dynamics. (anticipated length: 3-6 sentences)

6. Equation (B4), please add subscript 'i' to y-predicted

(AC) We will revise as suggested.

---

## Referee Comment (RC2) · Anonymous Referee #2 · 8 Feb 2021

The manuscript proposes a technique for filling stage or discharge measurement gaps using time-laps imagery from a single camera.

While the manuscript is well organized and the analyses refer to a good data set, I am skeptical on the aim of the paper, specifically on its usefulness.

As mentioned in the Introduction, Gauge-Cam system is an emerging topic and promising approach since it would allow to measure stage and velocity in challenging conditions were common approaches could fail. Time-laps imagery is able to provide stage and velocity measurements (or estimation), so I do not fully understand the reason to install a single camera just to fill gaps that, eventually, will be generated by the functioning of a traditional sensor. If the problem is to prevent possible service interruption, I would just include an additional similar sensor in the station, so if one does not work,

there will be time to substitute it without having gaps. The installation of a single camera is cheaper than traditional sensors, however it is not so easy and straightforward, and in any case implies costs. So, its implementation is justified if the aim is to provide additional information compared to the traditional sensor or to substitute it since it is cheaper. So, presently I do not see any practical usefulness of the proposed approach.

---

## Author Comment (AC2) · 9 Feb 2021

There are two layers to this. (1) There is a proliferation of ground-based cameras capturing water features for a variety of reasons. In our study, we partnered with the Platte Basin Timelapse project, a watershed documentary project that provided high quality images. Part of our goal was to test whether those images, freely available to the public, could be used to fill data gaps in streamflow records. (2) Because this paper demonstrates that these types of images can be used to fill data gaps, there is an opportunity to further explore the use of imagery to improve the reliability of stream monitoring networks. Imagery provides important context (as noted in the first sentence(s) of our abstract and introduction) and provides data validation that is not possible when simply installing a second sensor. Furthermore, while the images we used were from very

high quality cameras and set up by professional photographers, there are good quality, inexpensive (equal or less than the price of a second in-stream water level sensor) game cams that can capture images with extremely low maintenance requirements. In other words, imagery can provide a passive monitoring system with much lower cost for site visits (or potentially no cost, if the images are collected by another entity, for another purpose), with the potential benefits of data validation and gap filling based on information in the images. Thus, this study has very practical implications for improving stream monitoring sites and networks.